# Bivariate causal mixture model quantifies polygenic overlap between complex traits beyond genetic correlation

Oleksandr Frei [1], Dominic Holland[2,3,12], Olav B. Smeland [1,4,12], Alexey A. Shadrin[1], Chun Chieh Fan [2,5,6], Steffen Maeland[1], Kevin S. O'Connell [1], Yunpeng Wang[1,2,6], Srdjan Djurovic [7,8], Wesley K. Thompson[9,10], Ole A. Andreassen [1,4] & Anders M. Dale[2,3,6,11]

Accumulating evidence from genome wide association studies (GWAS) suggests an abundance of shared genetic influences among complex human traits and disorders, such as mental disorders. Here we introduce a statistical tool, MiXeR, which quantifies polygenic overlap irrespective of genetic correlation, using GWAS summary statistics. MiXeR results are presented as a Venn diagram of unique and shared polygenic components across traits. At 90% of SNP-heritability explained for each phenotype, MiXeR estimates that 8.3 K variants causally influence schizophrenia and 6.4 K influence bipolar disorder. Among these variants, 6.2 K are shared between the disorders, which have a high genetic correlation. Further, MiXeR uncovers polygenic overlap between schizophrenia and educational attainment. Despite a genetic correlation close to zero, the phenotypes share 8.3 K causal variants, while 2.5 K additional variants influence only educational attainment. By considering the polygenicity, discoverability and heritability of complex phenotypes, MiXeR analysis may improve our understanding of cross-trait genetic architectures.

[1] NORMENT, KG Jebsen Centre for Psychosis Research, Institute of Clinical Medicine, University of Oslo, 0424 Oslo, Norway. [2] Center for Multimodal Imaging and Genetics, University of California at San Diego, La Jolla, CA 92037, USA. [3] Department of Neurosciences, University of California, San Diego, La Jolla, CA 92093, USA. [4] Division of Mental Health and Addiction, Oslo University Hospital, 0407 Oslo, Norway. [5] Department of Cognitive Sciences, University of California at San Diego, La Jolla, CA 92093, USA. [6] Department of Radiology, University of California, San Diego, La Jolla, CA 92093, USA. [7] Department of Medical Genetics, Oslo University Hospital, 0424 Oslo, Norway. [8] NORMENT, KG Jebsen Centre for Psychosis Research, Department of Clinical Science, University of Bergen, 5020 Bergen, Norway. [9] Department of Family Medicine and Public Health, University of California, San Diego, La Jolla, CA 92093, USA. [10] Institute of Biological Psychiatry, Mental Health Center Sct. Hans, Capital Region of Denmark, Roskilde 4000, Denmark. [11] Department of Psychiatry, University of California, San Diego, La Jolla, CA 92093, USA. [12] These authors contributed equally: Dominic Holland, Olav B. Smeland. Correspondence and requests for materials should be addressed to O.F. (email: oleksandr.frei@gmail.com) or to A.M.D. (email: andersmdale@gmail.com)

In recent years, genome-wide association studies (GWASs) have successfully detected genetic variants associated with multiple complex human traits or disorders, providing important insights into human biology[1]. Understanding the degree to which complex human phenotypes share genetic influences is critical for identifying the etiology of phenotypic relationships, which can inform disease nosology, diagnostic practice, and improve drug development. Most human phenotypes are known to be influenced by multiple genetic variants, many of which are expected to influence more than one phenotypes (i.e., exhibit allelic pleiotropy)[2,3]. This has led to cross-trait analyzes, quantifying polygenic overlap, becoming a widespread endeavor in genetic research, made possible by the public availability of most GWAS summary statistics (*p*-values and *z*-scores)[4,5].

Currently, the prevailing measure to quantify polygenic overlap is genetic correlation. For a pair of traits, *polygenic overlap* refers to the fraction of genetic variants causally associated with both traits over the total number of causal variants across the two traits considered, while *genetic correlation* quantifies the correlation coefficient of additive genetic effects for the two traits. The sign of the correlation indicates whether the shared genetic effects predominantly have the same or the opposite effect directions. Available methods can quantify genetic correlation using raw genotypes[6,7] or GWAS summary statistics[8–11]. However, these methods report overall positive, negative, or no genetic correlation, but fail to capture mixtures of effect directions across shared genetic variants. This scenario is exemplified by the genetic relationship between schizophrenia and educational attainment. Despite consistent estimates of a

non-significant genetic correlation[12,13], many genome-wide significant loci are found to be jointly associated with both phenotypes[14]. Among 25 shared loci[15], 16 had effects in the opposite direction, while 9 had effects in the same direction. Thus, new statistical tools are needed to improve our understanding of the polygenic architecture of complex traits and their intricate relationships.

Here, we introduce a statistical tool (MiXeR), which quantifies polygenic overlap irrespective of genetic correlation between traits, using summary statistics from GWAS. To evaluate polygenic overlap between two traits, MiXeR estimates the total number of shared and trait-specific causal variants (i.e., variants with nonzero additive genetic effect on a trait). MiXeR bypasses the intrinsically difficult problem of detecting the exact location of causal variants, but rather aims at estimating their overall amount. MiXeR builds upon the univariate causal mixture model[16–19], which we extend to four bivariate normal distributions as illustrated in Fig. 1, with two causal components for variants specific to each trait, one causal component for variants affecting both traits, and a null component for variants with no effect on either trait. From the prior distribution of genetic effects, we derive the likelihood function of the observed signed test statistics (GWAS z-scores), incorporating effects of linkage disequilibrium (LD) structure, minor allele frequency (MAF), sample size, cryptic relationships, and sample overlap. The parameters of the mixture model are estimated from the summary statistics by direct optimization of the likelihood function.

We show in simulations that MiXeR provides accurate estimates of model parameters in the presence of realistic LD

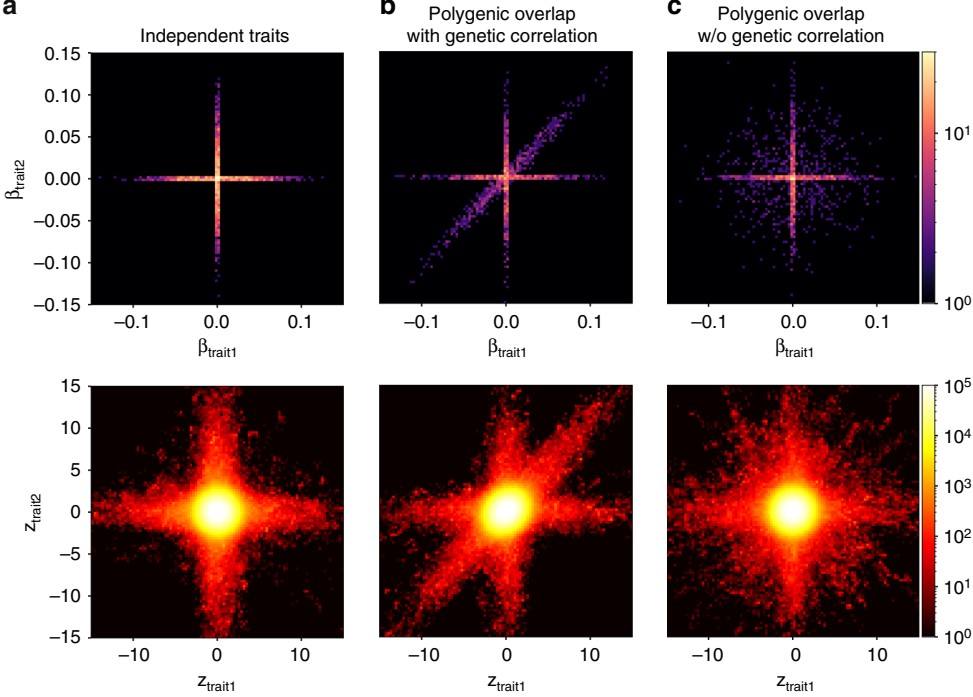

**Fig. 1** Components of the bivariate mixture in three scenarios of polygenic overlap. All figures are generated from synthetic data, where causal variants were drawn from the MiXeR model, the total polygenicity in each trait is set to 0.01%, SNP heritability is set to 0.4, GWAS $N = 100,000$. First column shows two traits where causal variants do not overlap. Second column adds a component of causal variants affecting both traits in the same (concordant) direction. Third column shows a scenario of polygenic overlap without genetic correlation. Top row shows simulated bivariate density of additive effects of allele substitution ($\beta_{1j}, \beta_{2j}$), the bottom row shows bivariate density of GWAS signed test statistics ($z_{1j}, z_{2j}$) for GWAS SNPs (genotyped or imputed). Due to linkage disequilibrium, GWAS-signed test statistic has substantially larger volume of SNPs associated with the phenotype. The aim of the MiXeR model is to infer distribution of causal effects (top row), using GWAS data (bottom row) as an input. Figures are generated on a regular grid of 100 × 100 bins, color histogram indicates $\log_{10}(N)$ where $N$ is the number of SNPs projected into a bin

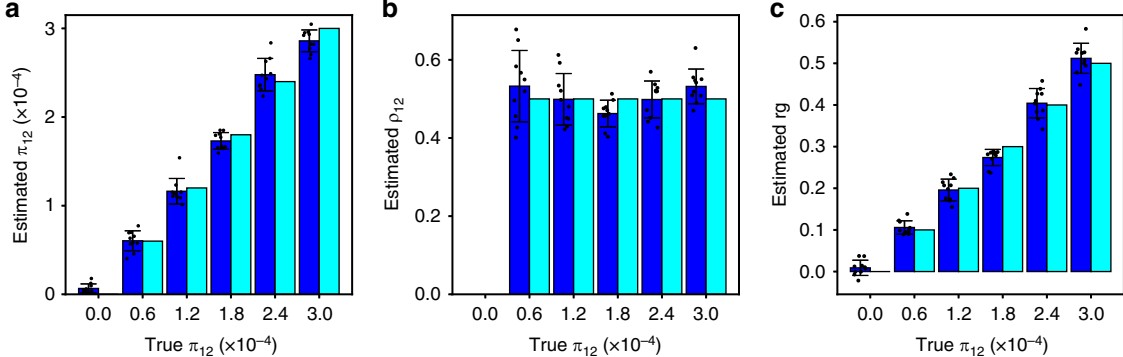

**Fig. 2** Selected simulations with bivariate model: **a** the estimates of polygenic overlap; **b** the estimates of correlation of the effect sizes in shared polygenic component; **c** the estimates of genetic correlation. The bars in blue indicate an average value of model estimates across ten simulation runs. The bars in cyan show true (simulated) parameters. Error bars represent standard deviation of the model estimate across ten simulation runs. Individual simulation runs are shown as dot points. Different bars correspond to levels of polygenic overlap: from zero (no overlap) to complete polygenic overlap. Simulated heritability is 0.4, simulated fraction of causal variants is 0.03% in both traits

structure. Using GWAS summary data, we quantify polygenic overlap of several psychiatric disorders, including schizophrenia and bipolar disorder, with educational attainment and human height, with large implications for understanding how genetic factors overlap between complex human phenotypes.

## Results

**Simulations studies.** In our first set of simulations, we generated synthetic GWAS data that follow model assumptions and assessed the validity of MiXeR estimates (polygenic overlap, $\pi_{12}$; correlation of effect sizes within the shared polygenic component, $\rho_{12}$, and genetic correlation, $r_g$) in the presence of a realistic LD structure (Fig. 2). We observed no bias in the estimates across a wide range of simulation scenarios (Supplementary Figs. 1–3), except for the scenario with low heritability ($h_2 = 0.1$) and high polygenicity ($\pi_1^u = 3 \times 10^{-3}$) in both traits, which represents an insufficiently powered GWAS study. In this scenario, MiXeR shows large variation among the estimates, and reports a non-zero polygenic overlap, when no such overlap exists. Standard errors estimated by the model are shown in Supplementary Table 1. Similarly, we show that univariate estimates of polygenicity and heritability are correct in all scenarios (Supplementary Fig. 4, Supplementary Table 2), except for the same scenario with low heritability and high polygenicity. In this scenario, a large variation among the polygenicity estimates suggests that the biases can be explained by truncated distribution of the errors, as the polygenicity parameters $\pi_1$, $\pi_2$, and $\pi_{12}$ are bound to be non-negative. Traits with low heritability should obtain correct MiXeR estimates when the GWAS are sufficiently powered.

In addition, we validated that the model accurately predicts GWAS quantile–quantile (Q–Q) plots (Supplementary Fig. 5) and detailed Q–Q plots with SNPs partitioned into disjoint groups according to MAF and LD score (Supplementary Fig. 6a, b). Detailed Q–Q plots show a stronger GWAS signal for SNPs with higher MAF and higher LD score. The model's prediction follows the same pattern, indicating that it correctly captures dependency of GWAS association statistics on MAF and LD score. To validate the accuracy of the predicted bivariate density, we generated conditional Q–Q plots (Supplementary Fig. 7), showing observed versus expected $-\log_{10} p$-values in the primary trait as a function of significance of association with a secondary trait at the level of $p \leq 0.1$, $p \leq 0.01$, and $p \leq 0.001$. We note that data Q–Q plots are closely reproduced by the model predictions across all $p$-value strata. Interestingly, scenarios without polygenic overlap show an enrichment, arising because GWAS $p$-values

depend on allele frequency and LD structure, though this effect is generally smaller than enrichment arising due to shared causal variants.

**Sensitivity analysis.** For sensitivity analysis, we conducted simulations with traits that have a shared pattern of differential enrichment of heritability across genomic categories[20], which are not accounted for by the MiXeR model. Simulations were informed by the enrichment pattern of schizophrenia[21], as estimated by stratified LD score regression[22] (Supplementary Fig. 8a). In the univariate analysis, polygenicity was underestimated by about 20% (Supplementary Fig. 9), suggesting that the model likely groups adjacent causal variants together and interpret their combined effect as arising from a single variant. In the bivariate analysis, we observed a small upwards bias in the estimate of polygenic overlap (Supplementary Table 3), but it did not exceed 10% of the polygenicity across all sufficiently powered scenarios.

Another assumption in the MiXeR model is that effect sizes are independent of allele frequencies. To asses this assumption, we ran simulations with effect sizes drawn from the BayesS[19] model (see Online Methods). It characterizes MAF-dependent architecture in terms of a single parameter S, ranging from S = 0 (equivalent to the MiXeR assumptions) to S = −1 (equivalent to the LDSR assumption). We simulated three intermediate values, S = −0.25, S = −0.5, and S = −0.75, which cover the range of BayesS estimates observed for real GWAS data. The results (Supplementary Fig. 10, Supplementary Tables 2, 4) highlight certain biases in MiXeR parameter estimates: for the extreme case of S = −0.75, heritability ($h_2$), univariate polygenicity ($\pi_1^u$), and polygenicity of the shared genetic component ($\pi_{12}$) are underestimated by up to 25%; correlation of effect sizes ($\rho_{12}$) is overestimated by 25%; however, the genome-wide genetic correlation ($r_g$) appears to have no bias.

Finally, we ran simulations with an incomplete reference, and simulated phenotypes where causal variants were spread across our entire reference panel of $N = 11{,}015{,}833$ variants, but only a fraction (50, 25, or 12.5%) of the variants enter LD structure estimation and fit procedure. The results (Supplementary Table 5) show that the total number of causal SNPs, as well as the heritability, are estimated correctly, while the polygenicity parameter is different from the simulated value, because it reflects the fraction of all tagged causal variants with respect to the reference that went into LD structure estimation.

**Table 1 The results of cross-trait analysis with the MiXeR model for schizophrenia (SCZ), bipolar disorder (BIP), educational attainment (EDU) and height**

| Trait 1 | Trait 2 | $n_{12}$ (se) | $n_1$ (se) | $n_2$ (se) | $\rho_{12}$ (se) | rg (se) | $rg_{LDSR}$ (se) |
|---------|---------|-----------|----------|----------|-----------|---------|-------------|
| SCZ | BIP | 6.19 (0.99) | 2.10 (1.26) | 0.21 (0.44) | 0.853 (0.019) | 0.725 (0.071) | 0.725 (0.024) |
| SCZ | EDU | 8.29 (0.84) | 0.00 (0.04) | 2.54 (1.02) | 0.071 (0.015) | 0.062 (0.014) | 0.079 (0.022) |
| SCZ | Height | 0.83 (0.10) | 7.46 (0.87) | 2.29 (0.12) | −0.045 (0.060) | −0.007 (0.010) | −0.008 (0.019) |
| BIP | EDU | 5.72 (1.46) | 0.68 (1.16) | 5.11 (1.58) | 0.278 (0.051) | 0.191 (0.036) | 0.188 (0.023) |
| BIP | Height | 0.83 (0.11) | 5.57 (1.11) | 2.29 (0.13) | 0.001 (0.067) | 0.000 (0.013) | −0.014 (0.021) |
| EDU | Height | 1.76 (0.11) | 9.07 (0.58) | 1.37 (0.10) | 0.519 (0.040) | 0.157 (0.010) | 0.141 (0.012) |

Columns: $n_{12}$ – estimated number of shared causal variants, reported in thousands; $n_1$ ($n_2$)– estimated number of causal variants, unique to trait 1 (trait 2), reported in thousands; $\rho_{12}$ – correlation of effect sizes in shared polygenic component; rg – genetic correlation ($r_g = \rho_{12}\pi_{12}/\sqrt{\pi_1^U \pi_2^U}$, see Online Methods); $rg_{LDSR}$—estimate of genetic correlation from LD Score Regression. The number of variants ($n_{12}$, $n_1$, and $n_2$) are adjusted to explain 90% of heritability in the corresponding component. Parameters are fitted using approximately 1.1 M HapMap3 SNPs

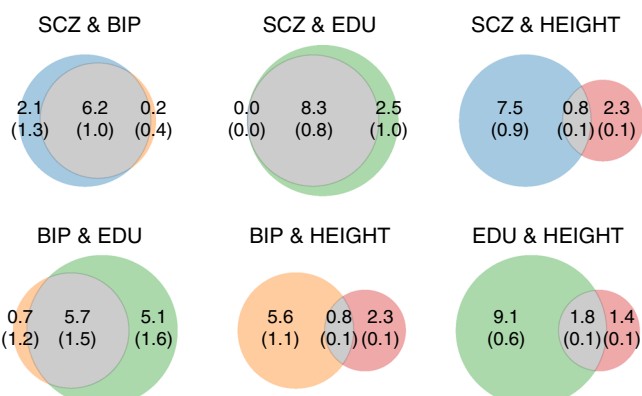

**Fig. 3** Venn diagrams of unique and shared polygenic components at the causal level, showing polygenic overlap (gray) between schizophrenia (SCZ, blue), bipolar disorder (BIP, orange), educational attainment (EDU, green), and height (red). The numbers indicate the estimated quantity of causal variants (in thousands) per component, explaining 90% of SNP heritability in each phenotype, followed by the standard error. The size of the circles reflects the degree of polygenicity

**GWAS summary statistics**. We applied MiXeR to summary statistics from GWAS of 14 phenotypes, including five psychiatric[21,23–26] and four autoimmune[27,28] diseases, four anthropomorphic traits[29–32], and educational attainment[12] (see Supplementary Table 6 for metadata about the studies).

MiXeR estimates of genetic correlation (Table 1; Supplementary Table 7) were generally consistent with those of cross-trait LD Score Regression[8], with the highest genetic correlation observed between schizophrenia and bipolar disorder. As expected, these disorders also exhibit substantial polygenic overlap, sharing 6.2 K out of 8.5 K causal variants involved. Here and below, the numbers of causal variants are reported as 22.6% of their total estimate, which jointly accounts for 90% of SNP heritability in each phenotype, to avoid extrapolating model parameters into the area of infinitesimally small effects (Supplementary Fig. 11).

Furthermore, MiXeR reveals important differences among traits with low genetic correlation, represented as Venn diagrams of shared and unique polygenic components (Fig. 3; Supplementary Figs. 12, 13a–g). For example, schizophrenia and educational attainment exhibit substantial polygenic overlap, sharing 8.3 K out of 10.8 K of causal variants involved. On the contrary, schizophrenia and height share only about 0.8 K out of 10.6 K causal variants. Educational attainment and height also show low polygenic overlap, sharing 1.8 K out of 12.3 K causal variants. Nevertheless, these traits have a high correlation of effect sizes

within the shared component, $\rho_{12} = 0.52$ (0.04), which at genome-wide level is observed as genetic correlation of rg = 0.16 (0.01) according to MiXeR, or rg = 0.14 (0.01) according to LDSR.

MiXeR estimates of the unique polygenic components provide insights into trait-specific genetic architectures. For example, schizophrenia has 2.1 K causal variants not shared with bipolar disorder, less than 0.1 K variants not shared with educational attainment, but as many as 7.5 K variants not shared with height. Also, for the other phenotypes, the number of trait-specific causal variants varies across different pairs of traits (Fig. 3).

Figure 4 and Supplementary Fig. 14a–g visualize the observed bivariate density of the GWAS-signed test statistics ($z_{1j}$, $z_{2j}$), the predicted density $\left(\hat{z}_{1j}, \hat{z}_{2j}\right)$ from the MiXeR model, and the estimated bivariate density of the additive causal effects ($\beta_{1j}$, $\beta_{2j}$) that underlie model predictions. Figure 4 gives real examples for the three different scenarios of polygenic overlap (genetically independent traits, polygenic overlap with and without genetic correlation, as shown in Fig. 1). Finally, we use conditional Q–Q plots[33,34] to compare the observed and predicted distributions of z-scores, and show that MiXeR-based prediction provides accurate estimates of the data Q–Q plots (Fig. 5), both in univariate and bivariate contexts.

## Discussion

MiXeR is a statistical method for cross-trait analysis of GWAS summary statistics, which enables a more complete quantification of polygenic overlap than provided by the LD score regression method[8]. In addition to genetic correlation, MiXeR estimates the total number of shared and trait-specific causal variants, providing new information about the genetic relationships between complex traits and disorders.

MiXeR extends cross-trait LD score regression[8] by incorporating a causal mixture model[16–19], thus relying on a biologically more plausible prior distribution of genetic effect sizes compared with the infinitesimal model[35,36]. We show that polygenicity, measured as a total number of causal variants, and discoverability[37], measured as variance of additive genetic effects across causal variants, have major implications for the future of GWAS discoveries (Supplementary Fig. 15).

Applying MiXeR to real phenotype data, we provide new insights into the genetic relationships between schizophrenia, bipolar disorder, educational attainment, and height. In line with the strong clinical relationship[38] between schizophrenia and bipolar disorder, and prior genetic studies[25,34,39,40], we find substantial polygenic overlap between the two disorders. Other studies have reported more substantial genetic differences between the disorders[40] (albeit with strong correlation[40,41]), likely because they did not specifically model the polygenic overlap. For

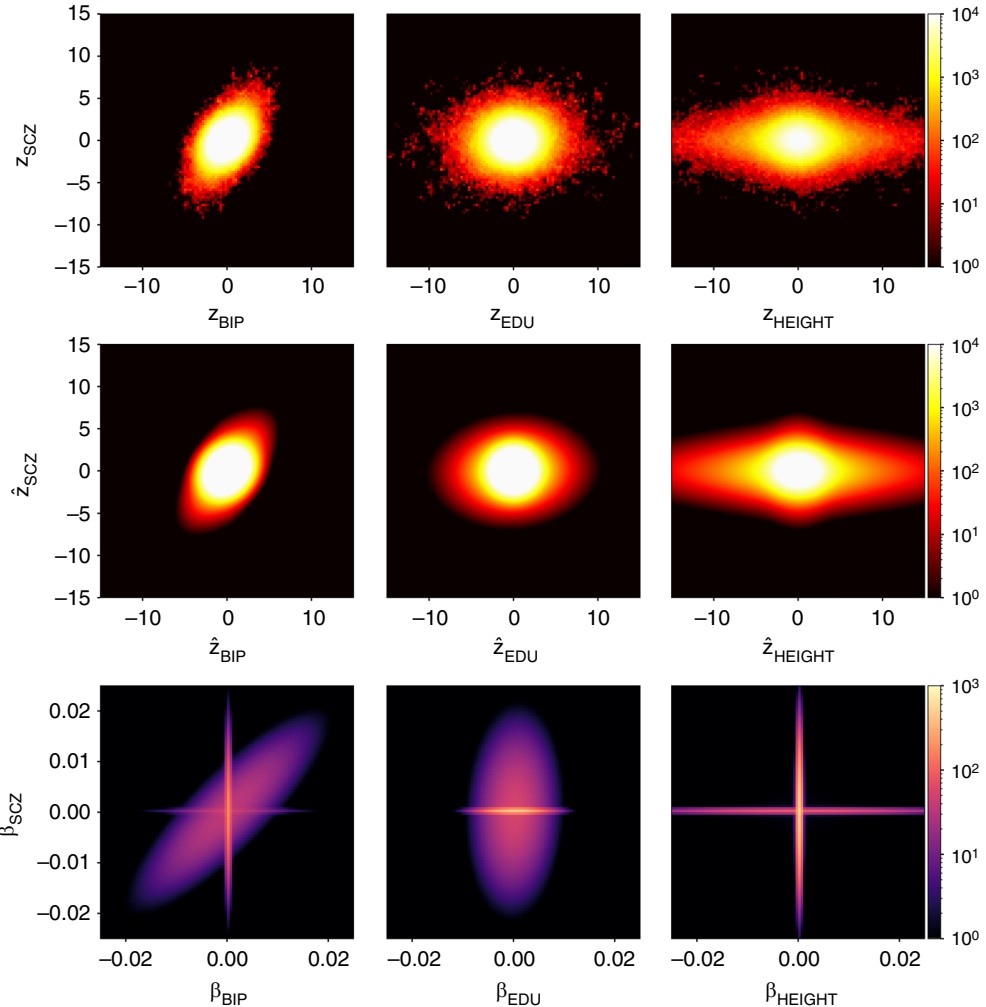

**Fig. 4** Top row shows bivariate density of the observed GWAS signed test statistics $(z_{1j}, z_{2j})$, middle row shows predicted density $(\hat{z}_{1j}, \hat{z}_{2j})$ from the MiXeR model. The bottom row shows estimated bivariate density of additive causal effects $(\beta_{1j}, \beta_{2j})$ that underlie model prediction. Three columns represent schizophrenia (SCZ) versus bipolar disorder (BIP), educational attainment (EDU), and height GWAS. Density is visualized using regular grid of 100 × 100 bins, color indicates $\log_{10}(N)$, where $N$ is the observed number (for the top row) or the expected number (for the middle and bottom rows) of SNPs projected into a bin

example, Ruderfer et al.[40] performed a combined GWAS of schizophrenia and bipolar disorder, and a differential GWAS of schizophrenia cases versus bipolar disorder cases. Our results indicate a higher polygenicity of schizophrenia than bipolar disorder, which is in line with the recent study by Bansal et al.[14], who highlighted two schizophrenia subtypes. Our results also indicate that both schizophrenia and bipolar disorder have a small fraction of causal variants conferring disorder-specific risk (Fig. 3). Identifying shared and disorder-specific genetic variants is a subject of our future research, as it could provide critical knowledge about the distinct genetic architectures underlying these psychiatric disorders. Moreover, we find that nearly all causal variants influencing schizophrenia risk also appear to influence educational attainment, despite a genetic correlation close to zero (Table 1). This is in line with recent studies demonstrating shared genetic loci between schizophrenia and educational attainment[15] and a strong genetic dependence between the phenotypes possibly related to different subtypes of schizophrenia[14]. In contrast, while 89% of genetic variants influencing bipolar disorder also appear to influence educational attainment, there is in this case a significant positive genome-

wide correlation of 0.191 (0.036), in agreement with the cross-trait LD score regression estimate of 0.188 (0.023) (Table 1; Supplementary Table 7).

We show that polygenicity is best expressed as a total number of causal variants (Supplementary Table 5). Previous studies presented it as a fraction, which is highly dependent on the reference panel used (1.1 M hapmap in ref. [18], or 484 K Affymetrix SNPs in ref. [19]). When expressed as a total number, our estimates of polygenicity are consistent with previously reported results[19] (Supplementary Table 8). In addition, we estimate that under the assumptions of the MiXeR model, only 5% of causal variants are needed to explain 50% of heritability, and 22.6% of causal variants are needed to explain 90% of heritability (Supplementary Fig. 11). These numbers are expected to be less dependent on modeling assumptions, because with finite GWAS samples it is not possible to distinguish small effects from truly null effects. The actual number of causal variants is, potentially, even higher, as our model tends to clump together variants if they are in high LD with each other (Supplementary Tables 3, 4).

Some existing methods can already uncover polygenic overlap in the absence of genetic correlation. For example, conjFDR

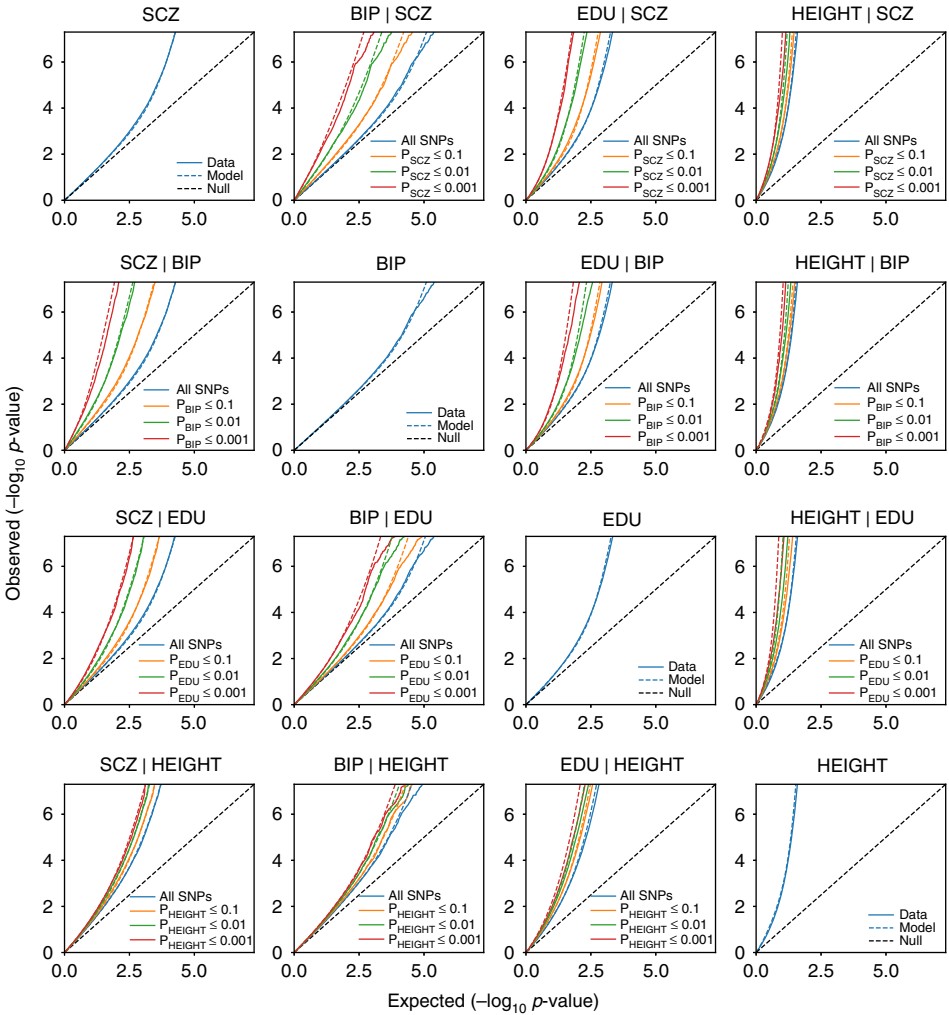

**Fig. 5** Conditional Q–Q plots of observed versus expected $-\log_{10}$ $p$-values in the primary trait as a function of significance of association with a secondary trait at the level of $p \leq 0.1$ (orange lines), $p \leq 0.01$ (green lines), $p \leq 0.001$ (red lines). Blue line indicates all SNPs. Dotted lines in blue, orange, green, and red indicate model predictions for each stratum. Black dotted line is the expected Q–Q plot under null (no SNPs associated with the phenotype). Points on the Q–Q plot are weighted according to LD structure, using $n = 64$ iterations of random pruning at LD threshold $r^2 = 0.1$

analysis[33,34] is a non-parametric model-free approach, which detects shared genetic loci regardless of their allelic effect directions, by prioritizing variants with strong associations across more than one GWAS[42]. Other methods, including gwas-pw[43] and HESS[44], also aim at detecting genomic loci jointly associated with two traits. MiXeR complements these methods by providing an easily interpretable high-level overview of the shared and unique genetic architectures underlying complex phenotypes. Among other notable methods for cross-trait analysis, the GenomicSEM[9] and MTAG[11] represent a multi-trait extension of the LD score regression. They can handle two or more traits at a time, but are based on the infinitesimal assumption, and quantify polygenic overlap using genetic correlation. For two-trait analysis, these methods are equivalent to LD score regression, thus we did not perform a formal comparison between GenomicSEM, MTAG, and MiXeR.

MiXeR has some notable advantages compared with the existing methods that implement causal mixture. First, our mathematical model for the likelihood term $p(z_j|\beta_j)$ is conceptually simpler and more flexible, resulting in unbiased estimates of model parameters across a wide range of simulation scenarios (Supplementary Figs. 1–3) and providing accurate

prediction of GWAS z-scores across varying ranges of MAF and LD (Supplementary Fig. 6a, b). Second, MiXeR implementation works well with a large reference of 10 M variants, while other methods have reduced the reference to 1.1 M HapMap SNPs (ref. [18]) or 484 K Affymetrix SNPs (ref. [19]). Finally, our model individually processes all SNPs, without grouping them into bins (ref. [16]).

MiXeR models causal effects as a single Gaussian component, while recent work[18,45] suggests that certain phenotypes, including height, require at least two causal components of small and large effects. We note that the MiXeR model still provides a good fit for SNPs not reaching the GWAS significance threshold (Supplementary Fig. 16) and shows deviations only toward the tail of the distribution. To further investigate the effects of model misspecification, we implemented right-censoring of genome-wide significant SNPs (see Online methods). The results (Supplementary Tables 7, 9) are consistent with our main analysis, except for height which received a lower estimate of heritability (65% instead of 70%), a slight increase in polygenicity, and increased polygenic overlap with other traits. We propose that for a better estimate of height's polygenicity, it would be beneficial to run MiXeR on a

residualized GWAS, after covarying association statistics for genotypes of all genome-wide significant SNPs.

Recent work suggests the importance of MAF- and LD-dependent genetic architectures[19,46], which are not directly modeled by MiXeR. Our simulations show certain biases in the estimates of polygenicity parameters ($\pi_1^u$ and $\pi_{12}$), which underestimate the true value by up to 25% in the case of extreme MAF-dependent architecture with S = −0.75 (Supplementary Fig. 10). However, these biases tend to cancel one another out when considering the relative size of the polygenic overlap ($\pi_{12}/\pi_1^u$ ratio). Also, on real data, most BayesS[19] estimates lay between S = −0.25 and S = −0.5, where the bias of the MiXeR estimates is in the order of 10% rather than 25%. On real data, we observe effects of MAF-dependent architectures by drawing Q–Q plots for subsets of SNPs (Supplementary Fig. 17a–g) partitioned into nine groups according to MAF and LD score, where the model tends to underestimate z-scores in low MAF bins. This effect, however, is quite subtle, and does not manifest itself on the overall Q–Q plots (Supplementary Fig. 16).

The MiXeR method requires large GWAS studies. Our recommendation is to apply MiXeR to studies with at least N = 50,000 participants, and inspect standard errors reported by MiXeR. In addition, MiXeR applies Bayesian information criterion (BIC) to compare causal mixture model versus the infinitesimal model, as shown in Supplementary Table 9. The cases where BIC selects the infinitesimal model indicate that the GWAS sample size is insufficient to reliably fit the polygenicity parameter. Generally, polygenicity estimation requires more GWAS power than heritability estimation, which can be visually explained by GWAS Q–Q plots (Supplementary Fig. 16): heritability is determined by the overall departure of the GWAS curve from the null line, while polygenicity is determined by its curvature, i.e., the point where the GWAS curve begins to bend upward from the null line, which is harder to estimate when GWAS signal is weak. This is captured by MiXeR standard errors, which show that individual parameters of the mixture model have lower estimation accuracy than their combinations—for example, relative errors for $\pi_1$ and $\sigma_\beta^2$ are larger than for the heritability estimate $h^2 \propto \pi_1 \sigma_\beta^2$, due to inversely-correlated errors (Supplementary Table 2). Despite these limitations, there is still a clear minimum on the energy landscape of cost function (Supplementary Figs. 18, 19, showing log-likelihood as a function of model parameters around the optimum).

In our future work, we plan to incorporate an additional Gaussian component to model small and large effects[18], and to explicitly account for MAF-dependent architectures[46]. Further extensions may account for differential enrichment for true associations across genomic annotations[20]. Another limitation to address is that the MiXeR model assumes similar LD structure among studies, and is not currently applicable for analysis across different ethnicities. We also aim to extend the MiXeR modeling framework to be used to improve power for discovery of shared and trait-specific SNPs by estimating the posterior effect size of SNPs associated with one trait given the test statistics in another trait, as well as for improving predictive power of polygenic risk scores.

In conclusion, MiXeR represents a useful addition to the toolbox for cross-trait GWAS analysis. By considering the intricate polygenic architectures of complex phenotypes, MiXeR allows for measures of polygenic overlap beyond genetic correlation. We expect this to lead to new insights into the pleiotropic nature of human genetic etiology.

## Methods
**Bivariate causal mixture model**. Consider a simple additive model of genetic effects, ignoring gene-environment interactions, epistasis and dominance effects.

Under these assumptions, the contribution of the genotype to the phenotype is modeled as a sum of individual contributions from genetic variants: $y_k = \sum_j g_{jk}\beta_j$, where $y_k$ is a quantitative phenotype or disease liability of $k$-th individual, $g_{jk}$ is 0, 1, 2-coded number of reference alleles for $j$-th variant, and $\beta_j$ is the additive genetic effect of allele substitution. We say that a genetic variant is causal for a trait if it has a non-zero effect on that trait ($\beta_j \neq 0$).

MiXeR builds upon the univariate causal mixture model[16],
$\beta_j \sim \pi_0 N(0,0) + \pi_1 N\left(0, \sigma_\beta^2\right)$, which assumes that only a small fraction ($\pi_1$) of variants have an effect on the trait, while the effect of the remaining variants is zero. For the mathematical convenience, we chose a Gaussian distribution for the non-null arm of the causal mixture. A drawback with the gaussian prior is that a large fraction of causal variants will have effect sizes close to zero. We would prefer to count a variant as causal only if it has a sufficiently large effect size, using for example a bi-modal prior distribution with probability mass separated from zero, but for such prior, it was not feasible to accurately model the effects of the LD structure.

In a joint analysis of two traits, we expect some variants to affect both traits; some variants to affect one trait but not the other; and most variants to have no effect on either trait. We assumed that for a given trait, all causal variants follow the same distribution of effect sizes, regardless of what effect a variant has on the other trait. Within the shared component, we model correlation of effect sizes, to account for genetically correlated traits. Based on these assumptions, MiXeR models additive genetic effects $\beta_{1j}$, $\beta_{2j}$ of variant $j$ on the two traits as a mixture of four bivariate Gaussian components (Fig. 1):

$$\left(\beta_{1j}, \beta_{2j}\right) \sim \pi_0 N(0,0) + \pi_1 N(0, \Sigma_1) + \pi_2 N(0, \Sigma_2) + \pi_{12} N(0, \Sigma_{12}), \quad (1)$$

$$\Sigma_1 = \begin{bmatrix} \sigma_1^2 & 0 \\ 0 & 0 \end{bmatrix}, \quad \Sigma_2 = \begin{bmatrix} 0 & 0 \\ 0 & \sigma_2^2 \end{bmatrix}, \quad \Sigma_{12} = \begin{bmatrix} \sigma_1^2 & \rho_{12}\sigma_1\sigma_2 \\ \rho_{12}\sigma_1\sigma_2 & \sigma_2^2 \end{bmatrix} \quad (2)$$

where $\pi_1$ and $\pi_2$ are weights of the unique components (variants with an effect on the first trait only, and on the second trait only); $\pi_{12}$ is a weighting of the component affecting both traits; and $\pi_0$ is a fraction of variants that are non-causal for both traits, $\pi_0 + \pi_1 + \pi_2 + \pi_{12} = 1$; $\sigma_1^2$ and $\sigma_2^2$ control expected magnitudes of per-variant effect sizes; and $\rho_{12}$ is the correlation coefficient of the effect sizes in the shared component. All parameters are assumed to be the same for all genetic variants.

The effects $\left(\hat{\beta}_{1j}, \hat{\beta}_{2j}\right)$ estimated by a GWAS, represent only proxies of the true causal effects ($\beta_{1j}$, $\beta_{2j}$), which are distorted by limited sample size (poor statistical power), cryptic relatedness within a GWAS sample, as well as LD between variants. To disentangle these effects, we derive the likelihood term for observed GWAS signed test statistics ($z_{1j}$, $z_{2j}$), incorporating effects of the LD structure (allelic correlation $r_{ij}$ between variants $i$ and $j$); heterozygosity $H_j = 2p_j(1 - p_j)$; where $p_j$ is the minor allele frequency of the $j$-th variant; the number of subjects genotyped per variant ($N_{1j}$ and $N_{2j}$); and variance distortion parameters $\sigma_{01}^2$, $\sigma_{02}^2$, and $\rho_0$. Specifically (see Supplementary Note 1),

$$\left(z_{1j}, z_{2j}\right) = \left(\delta_{1j}, \delta_{2j}\right) + N\left((0,0), \begin{bmatrix} \sigma_{01}^2 & \rho_0\sigma_{01}\sigma_{02} \\ \rho_0\sigma_{01}\sigma_{02} & \sigma_{02}^2 \end{bmatrix}\right), \quad (3)$$

$$\delta_{\cdot j} = \sqrt{N_{\cdot j}} \sum_i \sqrt{H_i} r_{ij} \beta_{\cdot j}$$

The nine parameters of the model ($\pi_1$, $\pi_2$, $\pi_{12}$, $\sigma_1^2$, $\sigma_2^2$, $\rho_{12}$, $\sigma_{01}^2$, $\sigma_{02}^2$, $\rho_0$) are fit by direct optimization of the weighted log likelihood, with standard errors estimated from the Observed Fisher's Information matrix.

Forcing $\pi_{12} = 1$ (so that $\pi_0 = \pi_1 = \pi_2 = 0$) reduces our model to an infinitesimal assumption that underlies cross-trait LD score regression[8]. Under this constraint, our model predicts that GWAS-signed test statistics follow a bivariate Gaussian distribution with zero mean and variance–covariance matrix

$$\Sigma_j = \ell_j \begin{bmatrix} N_{1j}\sigma_1^2 & \sqrt{N_{1j}N_{2j}}\rho_{12}\sigma_1\sigma_2 \\ \sqrt{N_{1j}N_{2j}}\rho_{12}\sigma_1\sigma_2 & N_{2j}\sigma_2^2 \end{bmatrix} + \begin{bmatrix} \sigma_{01}^2 & \rho_0\sigma_{01}\sigma_{02} \\ \rho_0\sigma_{01}\sigma_{02} & \sigma_{02}^2 \end{bmatrix},$$

i.e., $(z_{1j}, z_{2j}) \sim N(0, \Sigma_j)$, where $\ell_j = \sum_i H_i r_{ij}^2$ is the LD score (adjusted for heterozygosity). This model is consistent with cross-trait LD score regression, with expected chi-square statistics $E(z_{1j}^2)$, $E(z_{2j}^2)$, and cross-trait covariance $E(z_{1j}z_{2j})$ being proportional to the LD score of j-th SNP, and parameters $\rho_0$, $\sigma_{01}$, $\sigma_{02}$ playing the role of LD score regression intercepts[47]. The only distinction here is that we choose to model effect sizes that are independent of allele frequency, leading to the incorporation of $H_i$ into our model; this factor is absent from the LD score regression model due to the assumption there of effect sizes that are inversely proportional to $H_i$. Thus, MiXeR is a direct extension of cross-trait LD score regression, which relaxes the infinitesimal assumption.

**Model for bivariate distribution of GWAS z-scores**. We derive two models for GWAS z-scores, which we call "fast model" and "full model". The "fast model" is quicker to run, and we use it to perform an initial search in the space of the model's

parameters. The "full model" is slower but more accurate, and we use it for a final tuning of model estimates.

The "full model" for GWAS z-scores approximates $(z_{1j}, z_{2j})$ distribution of a given GWAS SNP as a mixture of K = 20,000 bivariate normal distributions, all having equal weight in the mixture. For each $k = 1, …, K$, we randomly draw the location of causal variants ($\pi_1 N$ causal variants specific to the first trait, $\pi_2 N$ specific to the second trait, and $\pi_{12} N$ shared causal variants, where $N$ denotes the total number of variants in the reference panel), and calculate the variance–covariance matrix $\Sigma'_{kj}$ from equation (3), using estimated LD $r^2$ correlations between the assumed causal variants and the GWAS SNP. Then

$$\binom{z_{1j}}{z_{2j}} = \binom{\delta_{1j}}{\delta_{2j}} + N\left((0,0), \begin{bmatrix} \sigma_{01}^2 & \rho_0 \sigma_{01} \sigma_{02} \\ \rho_0 \sigma_{01} \sigma_{02} & \sigma_{02}^2 \end{bmatrix}\right), \qquad (4)$$

$$\left(\delta_{1j}/\sqrt{N_{1j}}, \delta_{2j}/\sqrt{N_{2j}}\right) \sim \frac{1}{K} \sum_{k=1..K} N\left((0,0), \Sigma'_{kj}\right)$$

The "fast model" is derived from the method of moments (see Supplementary Note 1):

$$\left(\delta_{1j}/\sqrt{N_{1j}}, \delta_{2j}/\sqrt{N_{2j}}\right) \sim \left[(1-\pi'_{1j})N(0,0) + \pi'_{1j}N\left(0,\Sigma'_{1j}\right)\right] \oplus$$
$$\left[(1-\pi'_{2j})N(0,0) + \pi'_{2j}N\left(0,\Sigma'_{2j}\right)\right] \oplus \qquad (5)$$
$$\left[(1-\pi'_{12,j})N(0,0) + \pi'_{12,j}N\left(0,\Sigma'_{12,j}\right)\right],$$

where $\oplus$ denotes convolution of probabilistic distribution functions (so that right-hand side evaluates to a mixture of eight components), $\pi'_{cj} = \ell_j \pi_c / \eta_j$ is adjusted weight of mixture component ($c \in 1, 2, 12$); $\Sigma'_{cj} = \eta_j \Sigma_c$ is adjusted variance–covariance matrix; $\ell_j = \sum_i H_i r_{ij}^2$ is the LD score, adjusted for heterozygosity[48]; and $\eta_{cj} = \left(\pi_c \ell_j + (1-\pi_c)\frac{\sum_i H_i^2 r_{ij}^4}{\sum_i H_i r_{ij}^2}\right)$ can be interpreted as shape parameter that affects fourth and higher moments of the distribution. This model explains second moments $E\left[Z_{1j}^2\right]$, $E\left[Z_{1j}Z_{2j}\right]$, $E\left[Z_{2j}^2\right]$ and fourth moments $E\left[Z_{1j}^4\right]$, $E\left[Z_{1j}^2 Z_{2j}^2\right]$, $E\left[Z_{2j}^4\right]$ of z-score distribution, and forms a theoretical basis for the mixture model of sparse and ubiquitous effects[49,50]. Of interest is that the "fast model" involves the forth power of allelic correlation $r_{ij}^4$, which is directly proportional to kurtosis (measure of heavy tails) of z-score distribution.

**LD structure estimation.** To estimate the LD structure, we use 489 individuals from the 1000 Genome project[51] (phase 3 data), obtained from the LD score regression website[8,22,52]. In total, 14 individuals were excluded due to related-ness[53]. For simulations, LD scores were estimated from the actual genotypes that we use to produce synthetic GWAS summary statistics. LD $r^2$ coefficients were calculated using PLINK[54] with LD $r^2$ cutoff of 0.05 and fixed window size of 50,000 SNPs, corresponding on average to a window of 16 centimorgans. We deliberately chose a larger LD window compared with the LDSR-recommended window of 1 centimorgan, because the later appears to truncate a noticeable part of LD structure. At the same time, we did not observe an effect of using an unbiased estimate[55] of $r^2$, thus fall back to the standard Pearson correlation coefficient. We employ small integer compression[56] for efficient storage of the LD matrix.

**Fit procedure.** We fit the model by direct optimization of weighted log likelihood

$$F(\theta) = \sum_j w_j \log\left(pdf(z_j|\theta)\right), \qquad (6)$$

where $\theta = (\pi_1, \pi_2, \pi_{12}, \sigma_1^2, \sigma_2^2, \rho_{12}, \sigma_{01}^2, \sigma_{02}^2, \rho_0)$ is a vector of all parameters being optimized, and weights $w_j$ chosen by random pruning (64 iterations at LD $r^2$ 0.1). Optimization is done by the Nelder–Mead Simplex Method[57] as implemented in MATLAB's fminsearch. First, we fit univariate parameters separately for each trait, i.e., $\pi_1^u, \sigma_1^2, \sigma_{01}^2$ for the first trait, and similarly for the second trait. Univariate fit employs a sequence of optimizations to ensure robust convergence: first, we use the "fast model" under constraint $\pi_1^u = 1$ to find $\sigma_{1,inf}^2$ and to initialize $\sigma_{01}^2$; second, we use constraint $\pi_1^u \sigma_1^2 = \sigma_{1,inf}^2$ to find initial values of $\pi_1^u$ and $\sigma_1^2$, again with the "fast model". Finally, we use the "full model" and unconstrained optimization to jointly fit $\pi_1^u, \sigma_1^2, \sigma_{01}^2$ parameters. The same procedure is repeated for the second trait, to find $\pi_2^u, \sigma_2^2, \sigma_{02}^2$. To improve convergence, we parametrize univariate log-likelihood as a function of $\log(\pi_1^u \sigma_1^2)$ and $\log(\pi_1^u/\sigma_1^2)$, which represent almost independent dimensions of the energy landscape. In bivariate optimization, we use the "fast model" and constraint $\pi_{12} = 1$ to estimate $r_g$ and $\rho_0$. Then, we proceed with the "full model" optimization of the parameters specific to the bivariate model ($\pi_{12}, \rho_{12}$), constraining all other parameters to their univariate estimates, and also constraining $r_g$ and $\rho_0$ to the estimates from the infinitesimal model. The additional analysis (Supplementary Tables 7, 9) uses right-censoring[58] of z-scores exceeding $z_t = 5.45$, by using cumulative distribution function[59] in the

log likelihood:

$$F(\theta) = \sum_{j:|z_j| \leq z_t} w_j \log\left(pdf(z_j|\theta)\right) + \sum_{j:|z_j| > z_t} w_j \log\left(cdf(z_{max}|\theta)\right) \qquad (7)$$

**Standard error estimation.** We estimate standard errors of all parameters from the observed Fisher's information, based on the "fast model". It is known from the likelihood optimization theory that the observed Fisher's information may not be suitable for a parameter near its boundary, which is applicable to the mixture weights $\pi_1, \pi_2, \pi_{12}$, and the correlation of effect sizes $\rho_{12}$. To mitigate this problem, we apply transformations— MATLAB's logit () for $\pi_1, \pi_2, \pi_{12}$, exp() for $\sigma_1^2, \sigma_2^2, \sigma_{01}^2, \sigma_{02}^2$, and erf() for $\rho_0, \rho_{12}$, and estimated a variance–covariance matrix of errors in the transformed parameter space. We validated that our estimates based on the observed Fisher's information are in good agreement with block jack-knife estimates. To estimate standard errors for a function of the parameters, such as $r_g$ or $h^2$, we incorporate linear correlation among parameter errors in the transformed space. We sample $N = 1000$ realizations of the parameter vector, calculating the function (e.g., $r_g$ or $h^2$) on each of them, and report the standard deviations. In cases when joint hessian was not positive definite, we estimate marginal errors of fitted parameters.

**Akaike and Bayesian information criteria.** We apply standard formulas and estimate AIC $= 2k - 2F$ and BIC $= k \ln n - 2F$, where $F$ is the log-likelihood from Eqs. (6) or (7), $k$ is the number of parameters ($k = 2$ for an infinitesimal model, and $k = 3$ for causal mixture model), and $n = \sum_j w_j$ is the effective number of SNPs (the sum of weights across all SNPs used to fit the model).

**Large LD blocks.** The log-likelihood cost function and the Q–Q plots apply a weighting scheme to SNPs to 0avoid overcounting evidence from large LD blocks. As an alternative to weighting by inverse LD score, we chose to infer the weights by random pruning. This technique is a stochastic procedure which averages log-likelihood function across repeatedly selected subsets of variants, such that for each pair of variants $i, j$ in a subset $J$ the squared allelic correlation $r_{ij}^2$ falls below a certain threshold. Given $T$ iterations of random pruning the log-likelihood function can be calculated as follows:

$$F(\theta) = \frac{1}{T} \sum_{t=1}^{T} \sum_{j \in J_t} \log\left(pdf(z_j|\theta)\right) \qquad (8)$$

which is equivalent to weighted log-likelihood $F(\theta) = \sum_j w_j \log\left(pdf\left(z_j|\theta\right)\right)$ with weights $w_j = |\{t : j \in J_t\}|/T$, $|S|$ denotes cardinality of set S. Random pruning with stringent threshold $r^2 = 0.1$ justify independent modeling of the residuals in Eq. (3) across SNPs, which otherwise would be correlated.

**Heritability estimates.** In an additive model, SNP heritability is defined as a sum across all causal variants: $\sigma_\beta^2 \sum_{j:\beta_j \neq 0} 2p_j\left(1-p_j\right)$, which we approximate from an average heterozygosity of all variants in the reference: $\pi_1 H_{total} \sigma_\beta^2$, where $H_{total} = \sum_j 2p_j\left(1-p_j\right)$. To estimate the proportion of causal variants that explain a certain fraction of heritability (Supplementary Fig. 11), we randomly sample $N = 10,000$ causal effects from the reference, draw their effects $\beta_j$ from normal distribution, sort according to $\beta_j^2 p_j\left(1-p_j\right)$, and report the fraction of variants that cumulatively account for 90% of heritability.

**Genetic correlation.** Parameter $\rho_{12}$ in MiXeR defines the correlation of effect sizes within the shared polygenic component. Genome-wide genetic correlation, calculated across all SNPs, is related to $\rho_{12}$ by the following formula that involves polygenicity $\pi_1^u = \pi_1 + \pi_{12}$ and $\pi_2^u = \pi_2 + \pi_{12}$ of the traits, and polygenic over-lap $\pi_{12}$:

$$r_g = \rho_{12} \pi_{12}/\sqrt{\pi_1^u \pi_2^u} \qquad (9)$$

For traits with K-fold difference in polygenicity ($\pi_1^u = K\pi_2^u$), the formula predicts an upper bound on genome-wide genetic correlation: $r_g \leq \rho_{12}/\sqrt{K}$, where equality holds if causal variants of the less polygenic trait form a subset of the higher-polygenic trait.

**Quantile-quantile plots.** Univariate Q–Q plots and stratified Q–Q plots for the model were constructed from $pdf_j(z)$ density as defined by Eq. (3), given fitted parameters of the model and LD structure of j-th SNP, calculated across a fine grid of z-scores ranging from 0 to 38 with 0.05 step. We average $pdf_j(z)$ across 1% of randomly sampled SNPs, and numerically integrate the resulting probability density function to convert it into a cumulated distribution function. Error bars on data Q–Q plots represent the 95% binomial confidence interval $q \pm 1.96\sqrt{q(1-q)/n_{total}}$, where $q$ is the probability of observing a p-value as extreme as, or more extreme then the chosen p-value, and $n_{total}$ is the effective

number of SNPs after controlling for LD structure, which in our case was calculated as a sum of random pruning weights across all SNPs.

**GWAS power curves.** Causal mixture model can project the future of GWAS discoveries, by estimating proportion $S(N)$ of narrow-sense heritability captured by genome-wide significant SNPs at a given sample size $N$. The $S(N)$ is defined as follows:

$$S(N) = \frac{\sum_j \int_{z:|z|\geq z_t} C(z, N, j) dz}{\sum_j \int_z C(z, N, j) dz}, \tag{10}$$

where $z_t = 5.45$ gives z-score corresponding to the standard genome-wide significance threshold $5 \cdot 10^{-8}$, and $C(z, N, j) \equiv P(z, N, j) \cdot E(\delta^2|z, N, j)$ denotes a posterior effect size $E(\delta^2|z, N, j)$ of the non-centrality parameter $\delta^2$ for a GWAS SNP $j$, given certain z-score, multiplied by a prior probability of observing that z-score. Probability density function $P(z, N, j)$ is given by Eq. (4), and $E(\delta^2|z, N, j)$ can be calculated from the Bayesian rule. Thus, $C(z, N, j) = \int \delta^2 P(z|\delta) P(\delta, j) d\delta$, where $z|\delta \sim N(\delta, \sigma_0^2)$. Analytical expressions for $C(z, N, j)$ and $\int_{z:|z|\geq z_t} C(z, N, j) dz$ are given in the Supplementary Note 1.

**SNPs in the analysis.** To enable a direct comparison of our model with LD score regression, we use the same set of SNPs in our log-likelihood optimization, which consists of approx. 1.1 million variants, subset of 1000 Genomes and HapMap3[60], with MAF above 0.05, ambiguous SNPs excluded, imputation INFO above 0.9, MHC and other long-range LD regions excluded. Calculation of the LD structure, LD scores $\ell_j$ and shape parameter $\eta_j$ are based on 9,997,231 SNPs from 1000 Genomes Phase 3 data, downloaded from LD score regression website. In simulations we generate GWAS and estimate LD structure on a subset of 11,015,833 SNPs from 1000 Genomes Phase 3, with MAF above 0.002, call rate above 90%, excluding duplicated RS numbers; the fit procedure was constrained to ∼ 130 K GWAS SNPs, keeping only HapMap3 SNPs, and pruning SNPs at LD $r^2$ threshold of 0.1.

**LD score regression estimates.** For dichotomous phenotypes, we used an effective sample size of $N_{eff} = 4/(1/N_{case} + 1/N_{cont})$ to account for imbalanced numbers of cases and controls, both in MiXeR and in LD score regression. In addition, we ran LDSR using MiXeR MAF model (using `--per-allele` flags in LD score estimation), and show the results alongside with original LDSR estimates (Supplementary Tables 7, 9). For case/control phenotypes heritability is reported on the observed scale.

**Simulations.** In our simulations, we use a panel of $N = 100,000$ samples and 11,015,833 SNPs, generated by HapGen2[61] using 1000 Genomes[51] data to approximate the LD structure for European ancestry. To avoid relatedness across individuals, we run HapGen2 for small disjoint chunks of about 2900 SNPs at a time, 3920 chunks in total. The chunks were acting as additional recombination hotspots, causing certain changes in the distribution of the LD scores (Supplementary Fig. 20). However, the total amount of allelic correlation in the HapGen2 panel was still substantial, for example the median LD scores in the HapGen2 panel was 66.4, versus 63.5 in the 1000 Genomes panel, which makes the HapGen2 panel appropriate for our simulations.

We also validated that the HapGen2 panel shows no signatures of cryptic relatedness and sample stratification. The "`plink --pca`" analysis of the genotype matrix shows no signatures of sample stratification, as shown by the scatter plot of the first and second principal components (Supplementary Fig. 20). The "`plink --genome`" test found no related individuals (PI_HAT measure was below 0.1 for all pairs of individuals). We use a subset of 115,267 SNPs in the analysis, selected according to steps described in the PCA module of the Ricopili GWAS pipeline.

For each simulation run, we use PLINK to obtain GWAS summary statistics, including Wald's test z-score and p-value, of two synthesized quantitative phenotypes, with complete sample overlap between GWAS samples. Quantitative phenotype $y_k$ of k-th sample is calculated via a simple additive genetic model, $y_k = \sum_j g_{kj}\beta_j + \epsilon_k$, where $g_{kj}$ is the number of reference alleles for j-th SNP on k-th sample, $\beta_j$ is causal effect size, and $\epsilon$ is the residual vector drawn from normal distribution with zero mean and variance chosen in a way that sets heritability $h^2 = var(\mathbf{G}\beta)/var(y)$ to a predefined level.

For the simulations shown in Fig. 2, we draw effect sizes $(\beta_{1j}, \beta_{2j})$ from the four-component mixture model (Eq. (1)), varying polygenicity of each phenotype ($\pi_1^u = \pi_1 + \pi_{12}$ and $\pi_2^u = \pi_2 + \pi_{12}$), and polygenic overlap ($\pi_{12}$). We chose the total polygenicity in both traits to be $3 \times 10^{-3}$ or $3 \times 10^{-4}$ and include an additional scenario of uneven polygenicity ($\pi_1^u = 3 \times 10^{-3}$, $\pi_2^u = 3 \times 10^{-4}$). For each combination $\pi_1^u, \pi_2^u$ and $h_2$, we set polygenic overlap to be a fraction of total polygenicity $\pi_{12} = f\pi_1^u$, choosing the fraction $f$ from six equally spaced values (0.0 to 1.0 with a step of 0.2). Correlation of effect sizes $\rho_{12}$ set to 0.0 or 0.5. Heritability was set to 0.1, 0.4, or 0.7, which let us keep GWAS sample size constant ($N = 100,000$) because the distribution of GWAS z-scores depends on $N$ and $h_2$ only through their product, $h_2 \times N$ (thus, simulations with $N = 700,000$ and $h_2 = 0.1$ would be equivalent to our scenario with $N = 100,000$ and $h_2 = 0.7$). Finally, for

each combination of heritability, polygenicity, polygenic overlap, and correlation of effect sizes, we repeat simulations ten times.

For the simulations with differential enrichment, we simulate three levels of polygenicity (3 K, 30 K, and 300 K causal variants), three levels of heritability (0.1, 0.4, and 0.7), and for each combination, generate 20 pairs of genetically independent traits (except for having shared pattern of enrichment). To simulate the enrichment, we keep a constant variance of effect sizes across all SNPs, but modulate the probability of having causal variant proportionally to LDSR regression coefficient. We use `--per-allele` flags in LD score estimation to run simulations with the MiXeR MAF model.

For simulations with MAF-dependent architectures, we simulate effect sizes as follows:

$$\beta_j \sim \pi N\left(0, H_j^S \sigma_\beta^2\right) + (1 - \pi)N(0, 0) \tag{11}$$

Parameter $S = 0$ corresponds to the MiXeR MAF model, $S = -1$ corresponds to the LDSR MAF model. The same model is implemented in BayesS software[19], thus we choose parameter S from $-0.25$, $-0.50$, and $-0.75$, which corresponds to the range of BayesS estimates observed on real GWAS data.

## Data availability

The datasets analyzed during the current study are freely available for download from the following URLs: LD scores and reference panel derived from 1000 Genomes phase 3, https://data.broadinstitute.org/alkesgroup/LDSCORE/; Psychiatric Genomics Consortium (psychiatric disease), https://www.med.unc.edu/pgc/results-and-downloads/downloads; SSGAC (educational attainment), https://www.thessgac.org/data; GIANT Consortium, https://portals.broadinstitute.org/collaboration/giant; Early Growth Genetics Consortium, http://egg-consortium.org/birth-weight-2016.html; Rheumatoid arthritis, http://plaza.umin.ac.jp/~yokada/datasource/software.htm; International Inflammatory Bowel Disease Genetics Consortium, https://www.ibdgenetics.org/.

## Code availability

MiXeR software and a tutorial example on how to use it are available online (https://github.com/precimed/mixer, v0.9.1); PLINK software (v1.90b5.2, 64-bit, 9 Jan 2018), https://www.cog-genomics.org/plink2; LD score regression software, v1.0.0, https://github.com/bulik/ldsc; HapGen2 software, v2.2.0, http://mathgen.stats.ox.ac.uk/genetics_software; pipeline to harmonize GWAS summary statistics: https://github.com/precimed/python_convert (v0.9.1); pipeline to simulate synthetic GWAS data from genotypes: https://github.com/precimed/simu (v0.9.3).

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

## Acknowledgements

This work was supported by the Research Council of Norway (#223273, #225989, #248778) South-East Norway Health Authority (#2016-064, #2017-004), KG Jebsen Stiftelsen (#SKGJ-Med-008), and National Institutes of Health (R01MH100351, R01GM104400). The simulations were performed on resources provided by UNIN-ETT Sigma2—the National Infrastructure for High Performance Computing and Data Storage in Norway.

## Author contributions

Conceived and designed the study: A.M.D., O.A.A., and O.F.; method development: A.M.D., O.F., D.H., and A.A.S.; analysis and interpretation of results: O.F., D.H., A.A.S., O.B.S., C.C.F., Y.W., A.W., and S.D.; drafting the paper: O.F., O.A.A., and O.B.S.; revision and approval of the final paper: all authors.

## Additional information

**Competing interests:** Dr. Dale is a Founder of and holds equity in CorTechs Labs, Inc, and serves on its Scientific Advisory Board. He is a member of the Scientific Advisory Board of Human Longevity, Inc. and receives funding through research agreements

with General Electric Healthcare and Medtronic, Inc. The terms of these arrangements have been reviewed and approved by UCSD in accordance with its conflict of interest policies. Dr. Andreassen is a consultant for HealthLytix. The remaining authors have no competing interest.

