## [Peer Review File · Nature Communications]

Reviewers' comments:

Reviewer #1 (Remarks to the Author):

This paper describes a new method, MiXeR, which decomposes the pleiotropy between a pair of related phenotypes. It uses a bivariate mixture model to estimate the proportion of single-phenotype-specific component and pleiotropic component.

The method is straightforward. It extends the univariate mixture model – which has been widely used in the Bayesian framework to bivariate model. However, the method is not Bayesian – it maximizes the likelihood of z-scores in the similar context to LD score regression.

They performed simulations to show that their method works well for estimation of various parameters in a number of situations. Then they apply the method to the psychiatric disorders and educational phenotypes to interpret how a pair of phenotypes are related in the genomic structure.

I only have a few comments.

- Since this is a general method that can be applied to any phenotypes, I would suggest applying the method to autoimmune statistics (can be obtained from ImmunoBase etc). Autoimmune diseases are also related, and it will be interesting to see how the phenotypes are related in genomic structures. Also, morphological phenotypes (height, BMI, waist-hip-ratio) are interesting too. These statistics are well open to public. I observe that methods for cross-diseases often analyze more phenotypes than this paper has done (e.g. cross-trait LDSC)
- Method details in Supplementary Note is pretty interesting (the theory part up to page 4), and I would suggest moving quite a bit of contents to the main Methods, so that main Methods can be more self-contained.
- The authors note that for their method to work, the GWAS sample size has to be big. It would be good to have a guide line how and when the users can apply the method, and how they would know the results make sense.
- There are many typos: e.g. right-hand size π right-hand side
- Equation (5) in the main Methods: the variance of $(1-\pi)$ component has to be zero? Typo?

Reviewer #2 (Remarks to the Author):

The article introduces a new method MiXeR, which effectively extends the LD Score Regression framework to estimate not only the heritability and genetic correlation between pairs of traits, but also the fraction of causal variants for each trait and the fraction that have an effect on both traits. This further disentangling of shared genetic aetiology between pairs of phenotypes could provide extremely useful insights to motivate further investigations that may lead to a better understanding of disease and co-morbidities. However, while the article and novel method is thus of potential value, I have a number of issues that I would like addressing before I could consider recommending the article for publication in Nature Communications.

Major issues:

The authors use the Hapgen2 software to perform simulations. My group has found that Hapgen2 appears to simulate closely related, rather than distantly related or unrelated, individuals (presently being verified / checked with Hapgen2 authors). Thus, while it is no fault of the present authors, please do check whether the data simulated here using Hapgen2 corresponds to 'unrelated' individuals (IBD / PCA checks etc), and if not then whether that has compromised the extensive simulation studies performed.

A maximum polygenicity fraction of 0.001 is tested, yet the literature indicates that the true fraction could be much higher. eg. 0.06 as estimated by Zeng, ..., Yang. 2018 (Nature Genetics). Since polygenicity appears to affect the results of MiXeR, it would be good to get an idea of how the results may be affected if the polygenicity was an order higher.

MiXeR has poor performance under mis-specification of the relationship between effect sizes and MAF, to the extent that results could be highly misleading under the kind of mis-specification that would prevail if the LDSR assumptions were correct (including estimates of the polygenic overlap). As the authors will likely know, there has been a lively debate on this topic between the authors of LDSR and LDKA, and while those authors appear to agree that natural selection will act to mean that the effect sizes of low frequency risk alleles will be larger (the debate is about the size of this), the authors of the present study assume that natural selection does not act (effect sizes independent of MAF) without justification. Either the authors should provide a stronger justification for this model specification or else they should consider adjusting it.

While the conditional QQ plots are nice to see visually, it would be useful to include the QQ relating to conditioning at $P < 0.001$ on the trait itself (rather than only the other trait) to benchmark these departures from the null against a relevant 'maximum'. However, the utility of these conditional QQ plots is compromised by the finding of Supp.Fig.7 that there are substantial (the authors say minor) departures from the 'All SNPs' QQ even when there is no polygenic overlap. While it makes sense

that this can occur due to MAF and LD being correlated with P-values, it suggests that these conditional QQ plots should adjust for the effects of MAF and LD in order to have much use as a visual tool - misleading interpretations could result if not, because eg. traits under stronger selection may have larger departures from the null due to confounding by MAF, even if their polygenic overlap is the same. Thus, the authors should either urge stronger caution in interpretation of their conditional QQ plots or else attempt to adjust for the effects of MAF/LD so that the QQ plots show no/little departure under no polygenic overlap.

Minor issues:

The phrase “polygenic overlap independent of genetic correlation” sounds like an estimate conditional on the genetic correlation, when in fact it is an estimate of the overlap of causal variants irrespective of what the genetic correlation is. Perhaps the authors could make this distinction clearer.

I think the authors should do a little more to describe in lay terms how their method works and its assumptions - for example, whether priors for the expected polygenic overlap are specified and influence the results, what the prior distribution of genetic effects assumed is etc.

The results relating to the extremely high overlap between schizophrenia and bipolar, and the high correlation within the overlap, would have major implications if true. Therefore, it would be useful for the authors to explain this result in the context of other findings on the genetic basis of the two disorders, notably that other studies have typically found more substantial genetic differences between the disorders (albeit with strong correlation). Also, the estimates predict just 100 variants (of moderate effect) specific to bipolar, out of 7k - the readers will likely be interested in what these 100 variants are and whether they have function that supports the notion that they may be responsible for bipolar, as opposed to schizophrenia, in individuals (likewise the 2k variants specific to schizophrenia). Does MiXeR offer any way to get at the overlapping and non-overlapping variants? (even if not precisely)

At the start of the Results, the authors highlight some biases in MiXeR estimates - but they should also highlight the fact that (based on Supp. Fig 1) for low heritability traits, MiXeR often estimates polygenic overlap when no such overlap exists.

Why is it that the fraction of SNPs explaining 90% of the heritability (ie. 22.6%) is the same for all traits? (Supp. Fig.11). Surely their different genetic architectures would affect this figure? And doesn't the focus on these 22.6% of SNPs simply indicate that the authors should modify their prior for genetic effects to reflect that they are only interested in 'causal variants' to the extent that they have a sufficiently large effect on the trait?

The MiXeR model tends to clump variants together that are correlated with each other I presume, not 'located too close to each other' as stated.

In the inequality under equation (9), the \sqrt{k} should be \sqrt{K} .

The claim made above formula (10) is an interesting one - it would be nice to see whether MiXeR can indeed predict future GWAS discoveries well by using past iterations of GWASs of different sample size on a trait (as is available for numerous traits). I suggest this as optional revision, but it would be nice addition to the paper (not just in the Methods section).

Equation (10) includes an empty upper limit for the integration that was not added (repeated in the paragraph below).

The English in the Methods section should be checked more thoroughly as there are many spelling and grammatical errors.

Dear Dr. Trenkmann,

We thank you for the opportunity to revise our paper “Bivariate causal mixture model quantifies polygenic overlap between complex traits beyond genetic correlation”. The feedback from the reviewers has been helpful in improving the paper, and we have now revised the paper in accordance with their comments. We have uploaded a major revision, with substantial additions both to the main text and to the supplementary material. We also include a separate letter with full point-by-point response to all of the concerns raised by the reviewers.

We include in the submission Editorial Policy Checklist, Reporting Summary and Software Submission Checklist. Also, we updated Data Availability Statement in the manuscript according to the Nature Communication guidelines.

Additionally, we would like to comment on your following request:

>We would also request that you include formal comparisons with other existing methods to support your claim that “MiXeR is a novel method for cross-trait analysis of GWAS summary statistics, which enables a more complete quantification of polygenic overlap than provided by other existing tools¹⁻⁴”.

We thank you for drawing our attention to this statement, and we agree that formal comparisons between MiXeR and other available cross-trait analytical tools are warranted. In our original submission we had already included an extensive comparison between MiXeR and LDSR¹ (Supplementary Table 7), which is now extended with additional phenotypes as requested by the Reviewer #1. Moreover, we summarize differences between MiXeR and other cross-trait analytical tools in the methodology in the Discussion section, paragraph 5. Note that the MiXeR model works with two traits at a time, while two of the comparable methods (GenomicSEM² and MTAG⁴) represent multi-trait extensions to LDSR. Both GenomicSEM and MTAG are equivalent to LD score regression when they are applied to two traits. Hence, the MiXeR tool cannot be directly adequately compared with MTAG or GenomicSEM beyond our comparison with the LDSR method. A third relevant method, HESS³, focuses on locus discovery. MiXeR on the other hand aims at describing the genetic architecture of the two traits. Given these differences it is therefore impractical to provide a formal comparison between MiXeR and GenomicSEM, HESS and MTAG. However, we added a formal comparison between MiXeR and BayesS⁵ in the Supplementary Table 8, to evaluate the consistency between the estimates of polygenicity from these two methods. We hope that our formal comparison between LDSR, BayesS and MiXeR addresses your request.

We revised the first paragraph of the Discussion section and reduced our claim to “MiXeR enables more complete quantification of polygenic overlap than provided by the LDSR method”, i.e. replace “other existing tools” with “LDSR”. In the abstract, we replace “MiXeR provides more complete quantification of shared genetic architecture than offered by other available tools” with “MiXeR analysis may improve our understanding of cross-trait genetic architectures”.

Please let us know if you have further comments. We are looking forward to your response.

Kind regards,

Oleksandr Frei

References

1. Bulik-Sullivan, B. *et al.* An atlas of genetic correlations across human diseases and traits. *Nat Genet* **47**, 1236-41 (2015).
2. Verhulst, B., Maes, H.H. & Neale, M.C. GW-SEM: A Statistical Package to Conduct Genome-Wide Structural Equation Modeling. *Behav Genet* **47**, 345-359 (2017).
3. Shi, H., Mancuso, N., Spendlove, S. & Pasaniuc, B. Local Genetic Correlation Gives Insights into the Shared Genetic Architecture of Complex Traits. *Am J Hum Genet* **101**, 737-751 (2017).
4. Turley, P. *et al.* Multi-trait analysis of genome-wide association summary statistics using MTAG. *Nature Genetics* **50**, 229-237 (2018).
5. Zeng, J. *et al.* Signatures of negative selection in the genetic architecture of human complex traits. *Nature Genetics* **50**, 746-753 (2018).

> *Reviewers' comments:*

> *Reviewer #1 (Remarks to the Author):*

> *This paper describes a new method, MiXeR, which decomposes the pleiotropy between a pair of related phenotypes. It uses a bivariate mixture model to estimate the proportion of single-phenotype-specific component and pleiotropic component.*

> *The method is straightforward. It extends the univariate mixture model – which has been widely used in the Bayesian framework to bivariate model. However, the method is not Bayesian – it maximizes the likelihood of z-scores in the similar context to LD score regression.*

> *They performed simulations to show that their method works well for estimation of various parameters in a number of situations. Then they apply the method to the psychiatric disorders and educational phenotypes to interpret how a pair of phenotypes are related in the genomic structure.*

RESPONSE: We thank the reviewer for the positive comments and finding our paper of high interest.

> *I only have a few comments.*

> *-Since this is a general method that can be applied to any phenotypes, I would suggest applying the method to autoimmune statistics (can be obtained from ImmunoBase etc). Autoimmune diseases are also related, and it will be interesting to see how the phenotypes are related in genomic structures. Also, morphological phenotypes (height, BMI, waist-hip-ratio) are interesting too. These statistics are well open to public. I observe that methods for cross-diseases often analyze more phenotypes than this paper has done (e.g. cross-trait LDSC)*

RESPONSE: We have extended our analysis to four autoimmune diseases: rheumatoid arthritis (RA), inflammatory bowel disease (IBD), ulcerative colitis (UC), Crohn's disease (CD); and three anthropomorphic traits: birth weight, waste hip ratio (WHR), body mass index (BMI), in addition to height which was already covered in our previous analysis. The new results are presented in:

- Supplementary Figures 13h, 13i – Venn diagrams and conditional QQ plots
- Supplementary Figures 14h, 14i – Observed and estimated bivariate density of z scores
- Supplementary Figure 15 – projected power of future GWAS
- Supplementary Figure 16 – univariate QQ plots
- Supplementary Figures from 17h to 16n – partitioned QQ plots into MAF and LD bins
- Supplementary Table 6 – meta-data about GWAS studies
- Supplementary Tables 7, 9 – univariate and bivariate parameter estimates

The new results look reasonable from our point of view. However, we did not have an opportunity to discuss them in detail with domain experts, and we would like to point out that the MiXeR methodology was primarily developed for phenotypes of high polygenicity, while the autoimmune diseases investigated have low polygenicities. Another important limitation for the analysis of autoimmune diseases is that MiXeR excludes the MHC region, given the complex patterns of the LD structure, despite its major role in autoimmune diseases. Also, MiXeR requires genome-wide summary statistics while many GWAS studies for auto-immune diseases used ImmunoChip. We included all new results as Supplementary material, to keep the focus on the methodology behind MiXeR and on validation with synthetic data.

>- *Method details in Supplementary Note is pretty interesting (the theory part up to page 4), and I would suggest moving quite a bit of contents to the main Methods, so that main Methods can be more self-contained.*

RESPONSE. We thank the reviewer for the positive comment. The mathematical details behind our “fast model” appear to be quite technical, and we anticipate that due to diverse background and expertise many readers of the *Nature Communication* might not benefit from having it in the main paper, thus we tentatively keep it as a Supplementary Note.

>- *The authors note that for their method to work, the GWAS sample size has to be big. It would be good to have a guide line how and when the users can apply the method, and how they would know the results make sense.*

RESPONSE. The required GWAS sample size depends on heritability and polygenicity of a trait. Recommended sample size is to have at least 50,000 GWAS participants, as we have specified in the discussion section. To further aid our users we now implement Akaike (AIC) and Bayesian (BIC) information criteria and incorporate it into the MiXeR model to validate whether causal mixture has a better fit than the “infinitesimal” model. We include the following recommendation in the Discussion section: “The cases where BIC selects the infinitesimal model indicate that GWAS sample size is insufficient to reliably fit the polygenicity parameter”. Applying this approach to the real phenotypes (Supplementary Table 9) highlight two traits from our supplementary analysis (MDD and ASD), which are sufficiently powered according to the AIC criteria, but not according to BIC criteria (which is expected as BIC is always more conservative than AIC). The reason is that MDD has very low heritability and high polygenicity, and ASD has low GWAS sample size. This is in line with Supplementary Figure 16, where QQ plots for MDD and ASD have very wide confidence intervals, largely overlapping with the null GWAS curve.

>- *There are many typos: e.g. right-hand size vs right-hand side*

RESPONSE. We thank the reviewer for pointing this out. We corrected this typo and have thoroughly checked the manuscript and corrected many spelling and grammatical errors.

>- *Equation (5) in the main Methods: the variance of (1-pi) component has to be zero? Typo?*

RESPONSE. We thank the reviewer for pointing out this mistake. This typo is now corrected.

We would also like to highlight changes that we made in response to the other reviewer:

1. We validated HapGen2 reference panel with PCA/IBD checks, and summarized the results in a new Supplementary Figure 20
2. We compared polygenicity estimates between MiXeR and the BayesS¹ software in a new Supplementary Table 8
3. We extended our sensitivity analysis for MAF-dependent architectures and provide new extensive results in Supplementary Figure 10 and Supplementary Table 4.
4. We found and fixed a minor detail in the fit procedure that lead to the biases in π_{12} and ρ_{12} estimates in the presence of genetic correlation (Supplementary Figure 1b and

2b). This is now corrected, and all relevant figures and tables were updated accordingly, including main text figures 2,3,4,5.

5. In the analysis of real GWAS data we've incorrectly multiplied our polygenicity estimates to the size of our Hapgen2 reference from simulations (11,015,833 SNPs), instead of multiplying by the size of LDSR reference (9,997,231 SNPs). We have now fixed this, which reduce the number of causal variants by roughly 10% but did not conceptually change any of our results.

References

1. Zeng, J. *et al.* Signatures of negative selection in the genetic architecture of human complex traits. *Nature Genetics* **50**, 746-753 (2018).

> Reviewer #2 (Remarks to the Author):

>The article introduces a new method MiXeR, which effectively extends the LD Score Regression framework to estimate not only the heritability and genetic correlation between pairs of traits, but also the fraction of causal variants for each trait and the fraction that have an effect on both traits. This further disentangling of shared genetic aetiology between pairs of phenotypes could provide extremely useful insights to motivate further investigations that may lead to a better understanding of disease and co-morbidities. However, while the article and novel method is thus of potential value, I have a number of issues that I would like addressing before I could consider recommending the article for publication in Nature Communications.

RESPONSE: We thank the reviewer for the positive comments and finding our paper of high interest.

> Major issues:

>The authors use the Hapgen2 software to perform simulations. My group has found that Hapgen2 appears to simulate closely related, rather than distantly related or unrelated, individuals (presently being verified / checked with Hapgen2 authors). Thus, while it is no fault of the present authors, please do check whether the data simulated here using Hapgen2 corresponds to 'unrelated' individuals (IBD / PCA checks etc), and if not then whether that has compromised the extensive simulation studies performed.

RESPONSE. We thank the reviewer for pointing this out. We ran additional analysis (IBD/PCA checks) and found no signatures of cryptic relatedness or sample stratification in our Hapgen2 panel. The results are presented in the new Supplementary Figure 20, and described in "Online Methods. Simulations" section, which we copy below:

"The "plink --pca" analysis of the genotype matrix shows no signatures of sample stratification, as shown by the scatter plot of the first and second principal components (Supplementary Figure 20). The "plink --genome" test found no related individuals ("pihat < 0.1" for all pairs of individuals). We use a subset of 115,267 SNPs in the analysis, selected according to steps described in PCA module of the Ricopili GWAS pipeline. To avoid relatedness across individuals we run HapGen2 in small disjoint chunks of about 2900 SNPs at a time, 3920 chunks in total. The chunks were acting as additional "recombination hotspots", causing small changes in the distribution of the LD scores (Supplementary Figure 20). However, the amount of allelic correlation in HapGen2 panel was still substantial, for example the median LD scores in HapGen2 panel was 66.4, versus 63.5 in 1000 Genomes panel, which makes HapGen2 panel appropriate for our simulations."

>A maximum polygenicity fraction of 0.001 is tested, yet the literature indicates that the true fraction could be much higher. eg. 0.06 as estimated by Zeng, ..., Yang. 2018 (Nature Genetics). Since polygenicity appears to affect the results of MiXeR, it would be good to get an idea of how the results may be affected if the polygenicity was an order higher.

RESPONSE. Polygenicity of 0.06 from the BayesS method (Zeng, ... Yang 2018, Nature Genetics)¹ is estimated with respect to a reference of 483,634 SNPs, and is equivalent to polygenicity of 0.0029 using the MiXeR reference of 9,997,231 SNPs. The latter is within the

range that we covered in simulations (highest polygenicity tested was 0.003). In our original submission we ran simulations with incomplete reference (Supplementary Table 5), which highlight an important observation: causal mixture model correctly estimates the total number of SNPs, while polygenicity parameter should be interpreted only in the context of the reference set of SNPs. To further clarify this observation, we now include an additional table (Supplementary Table 8), which compares polygenicity estimates from BayesS and MiXeR. The results show that both methods give consistent estimates for the total number of causal variants; discrepancies up to a factor of 2 are expected, given the difference in input data (summary stats from consortia versus UKB), difference in methodology behind BayesS and MiXeR, and 20 times larger reference in MiXeR analysis.

During this analysis (Supplementary Table 8) we discovered that for the real GWAS data we've incorrectly multiplied our polygenicity estimates by the size of our Hapgen2 simulation reference (N=11,015,833), instead of multiplying by the size of LDSR reference (N=9,997,231). We have now fixed this, which reduce all numbers of causal variants by approximately 10% but did not conceptually change any of our results.

>MiXeR has poor performance under mis-specification of the relationship between effect sizes and MAF, to the extent that results could be highly misleading under the kind of mis-specification that would prevail if the LDSR assumptions were correct (including estimates of the polygenic overlap). As the authors will likely know, there has been a lively debate on this topic between the authors of LDSR and LDAK, and while those authors appear to agree that natural selection will act to mean that the effect sizes of low frequency risk alleles will be larger (the debate is about the size of this), the authors of the present study assume that natural selection does not act (effect sizes independent of MAF) without justification. Either the authors should provide a stronger justification for this model specification or else they should consider adjusting it.

RESPONSE. We thank the reviewer for pointing this out. We have run an extensive set of additional simulations with MAF-dependent genetic architecture, summarized in a new Supplementary Figure 10, and described the results in the "Sensitivity analysis" and in the "Discussion" sections:

"... we ran simulations with effect sizes drawn from the BayesS model (see Online Methods). It characterizes MAF-dependent architecture in terms of a single parameter S , ranging from $S=0$ (equivalent to the MiXeR assumptions) to $S=-1$ (equivalent to the LDSR assumption). We simulated three intermediate values, $S=-0.25$, $S=-0.5$, and $S=-0.75$, which cover the entire range of BayesS estimates observed for the real GWAS data. The results (Supplementary Figure 10, Supplementary Tables 2, 4) highlight certain biases in MiXeR parameter estimates: for the extreme case of $S=-0.75$, heritability (h^2), univariate polygenicity (π_1^u), and polygenicity of the shared genetic component (π_{12}) are underestimated by up to 25%; correlation of effect sizes (ρ_{12}) is overestimated by 25%; the genome-wide genetic correlation (r_g) appears to have no bias."

"... these biases tend to cancel one another out when considering the relative size of the polygenic overlap (π_{12}/π_1^u ratio). Also, on real data most BayesS estimates lay between $S=-0.25$ and $S=-0.5$, where the bias of the MiXeR estimates is in the order of 10% rather than 25%."

From our view these results justify our current MiXeR MAF model for the purpose of estimating polygenic overlap.

We have removed the obsolete supplementary material:

- [REMOVED] Supplementary Figure 10. Sensitivity analysis: polygenic overlap estimates under genomic enrichment and mis-specified MAF architecture
- [REMOVED] Supplementary Table 4. Sensitivity analysis: polygenic overlap estimates under genomic enrichment and mis-specified MAF architecture

which is superseded by the following material:

- [ADDED] Supplementary Figure 10. Sensitivity analysis for mis-specified MAF-dependent architecture (univariate and bivariate estimates)
- [ADDED] Supplementary Table 4. Sensitivity analysis: parameter estimates in bivariate analysis under mis-specified MAF architecture

>While the conditional QQ plots are nice to see visually, it would be useful to include the QQ relating to conditioning at $P < 0.001$ on the trait itself (rather than only the other trait) to benchmark these departures from the null against a relevant 'maximum'.

RESPONSE. A similar scenario is, to some extent, covered by new results on autoimmune diseases, which we now include in Supplementary Figure 13h. In the original GWAS study the IBD analysis basically combined CD and UC groups and treated both of them as cases, resulting in a substantial sample overlap between the studies. As a result, the " $p < 0.001$ " stratum in cross-trait analysis of IBD|CD, IBC|UC, CD|IBD, UC|IBD starts almost at $\log P = 2$ on the vertical axis. Similarly, an analysis at $P < 0.001$ on the trait itself will imply that the conditional QQ plot starts at $\log P = 3$ on the vertical axis.

>However, the utility of these conditional QQ plots is compromised by the finding of Supp.Fig.7 that there are substantial (the authors say minor) departures from the 'All SNPs' QQ even when there is no polygenic overlap. While it makes sense that this can occur due to MAF and LD being correlated with P-values, it suggests that these conditional QQ plots should adjust for the effects of MAF and LD in order to have much use as a visual tool - misleading interpretations could result if not, because eg. traits under stronger selection may have larger departures from the null due to confounding by MAF, even if their polygenic overlap is the same. Thus, the authors should either urge stronger caution in interpretation of their conditional QQ plots or else attempt to adjust for the effects of MAF/LD so that the QQ plots show no/little departure under no polygenic overlap.

RESPONSE. The purpose of the conditional QQ plots in our paper is to validate that the bivariate distribution of the p-values estimated by the MiXeR model closely reproduces the same distribution observed in GWAS, without making statements about pleiotropic enrichment or polygenic overlap. We have removed claims about polygenic overlap from all conditional QQ plots. We agree that the enrichment on the Suppl. Fig. 7 is not minor, and we have revised the text accordingly.

> Minor issues:

> The phrase "polygenic overlap independent of genetic correlation" sounds like an estimate conditional on the genetic correlation, when in fact it is an estimate of the overlap of causal

variants irrespective of what the genetic correlation is. Perhaps the authors could make this distinction clearer.

RESPONSE. We revised the paper to avoid the term “independent of genetic correlation”.

> I think the authors should do a little more to describe in lay terms how their method works and its assumptions - for example, whether priors for the expected polygenic overlap are specified and influence the results, what the prior distribution of genetic effects assumed is etc.

RESPONSE. We revised the Online Methods section to explain for choice of the prior distribution of genetic effects used in the MiXeR model:

“For the mathematical convenience we choose a gaussian distribution for the non-null arm of the causal mixture. A drawback with the gaussian prior is that a large fraction of “causal” variants will have effect sizes close to zero. We would prefer to count a variant as causal only if it has a sufficiently large effect size, using for example a bi-modal distribution with probability mass separated from zero, but for such prior it was not feasible to model the effects of the LD structure.”

“In a joint analysis of two traits we assumed that for a given trait all causal variants follow the same distribution of effect sizes, regardless of what effect a variant has on the other trait. Within the shared component that we model correlation of effect sizes, to account for genetically correlated traits.”

The expected effects of the prior distribution on the results are now described in the Discussion section, including the effects from difference MAF-dependent architectures. A general overview of how the method works is available in the introduction: “From the prior distribution of genetic effects, we derive likelihood function of the observed signed test statistics (GWAS z-scores), incorporating effects of linkage disequilibrium (LD) structure, minor allele frequency, sample size, cryptic relationships, and sample overlap. The parameters of the mixture model are estimated from the summary statistics by direct optimization of the likelihood function”.

Also, in the introduction we clarify distinction between genetic overlap and genetic correlation: “For a pair of traits, genetic overlap refers to the proportion of genetic variants causally associated with both traits, while genetic correlation quantifies the correlation coefficient of additive genetic effects for the two traits.”

> The results relating to the extremely high overlap between schizophrenia and bipolar, and the high correlation within the overlap, would have major implications if true. Therefore, it would be useful for the authors to explain this result in the context of other findings on the genetic basis of the two disorders, notably that other studies have typically found more substantial genetic differences between the disorders (albeit with strong correlation).

RESPONSE. We thank reviewer for pointing this out, and revise the Discussion section:

“In line with the strong clinical relationship between schizophrenia and bipolar disorder, and prior genetic studies, we find substantial polygenic overlap between these two disorders. However, other studies have found more substantial genetic differences between the disorders (albeit with strong correlation), likely because they have not specifically modelled polygenic overlap, as for example in the Ruderfer et al study² which performed a combined GWAS of schizophrenia and bipolar disorder, and a differential GWAS of schizophrenia cases versus

bipolar disorder cases. Our results show higher polygenicity in schizophrenia, which is in line with Bansal et al³ that highlight two schizophrenia sub-types”.

> Also, the estimates predict just 100 variants (of moderate effect) specific to bipolar, out of 7k - the readers will likely be interested in what these 100 variants are and whether they have function that supports the notion that they may be responsible for bipolar, as opposed to schizophrenia, in individuals (likewise the 2k variants specific to schizophrenia). Does MiXeR offer any way to get at the overlapping and non-overlapping variants? (even if not precisely)

RESPONSE. We agree with the reviewer that discovery of specific schizophrenia and bipolar disorder markers is of high interest and is a prioritized effort for our research team. This has now been included in the Discussion section: “Identifying shared and disorder-specific genetic variants is a subject of our future research.”

The MiXeR model allows for posterior estimates of effect sizes at the level of GWAS markers (for example, see formulas (24), (26) from the Supplementary Note, which allows calculation of $E(\delta^2|z)$). We are currently extending this to the bivariate context, and also exploring the possibility to use causal mixture model for fine-mapping. However, it is too early to include these results in the current manuscript.

> At the start of the Results, the authors highlight some biases in MiXeR estimates - but they should also highlight the fact that (based on Supp. Fig 1) for low heritability traits, MiXeR often estimates polygenic overlap when no such overlap exists.

RESPONSE. We agree with the reviewer and have revised the “Results. Simulation studies” section and highlighted the fact that in certain cases MiXeR may find polygenic overlap when no such overlap exists. We also point out that traits with low heritability should still have correct estimates for sufficiently large GWAS. It has previously been shown that GWAS power depends on the product $N \cdot h^2$ (N – GWAS sample size, h^2 – heritability of the trait, ref⁴), so from our simulations it follows that GWAS of 400,000 individuals will be sufficient to apply MiXeR to traits with $h^2=0.1$. We highlight this in “Online Methods. Simulation” section (4th paragraph).

Additionally, we have included a note regarding biases of the MiXeR estimates that we reported originally. We have now found and fixed a minor detail in the fit procedure that lead to the biases in π_{12} and ρ_{12} estimates in the presence of genetic correlation (Supplementary Figure 1b and 2b). This is now corrected, and all relevant figures and tables were updated accordingly, including main text figures 2,3,4,5.

>Why is it that the fraction of SNPs explaining 90% of the heritability (ie. 22.6%) is the same for all traits? (Supp. Fig. 11). Surely their different genetic architectures would affect his figure? And doesn't the focus on these 22.6% of SNPs simply indicate that the authors should modify their prior for genetic effects to reflect that they are only interested in 'causal variants' to the extent that they have a sufficiently large effect on the trait?

RESPONSE. The purpose of 22.6% is, indeed, to count “causal variants” with sufficiently large effects. We have now clarified this in the “Online Methods. Bivariate causal mixture model” section, paragraph 2. The idea to focus on large effects has been previously suggested, for example, in ref⁵: “it may be preferable to compare genetic architecture across traits in terms of the number of susceptibility SNPs that may have meaningfully large effects, such as an odds-ratio of 1.01 or larger”.

A fixed fraction of SNPs explaining 90% of heritability (the 22.6%) is a direct consequence of our modeling assumptions, including the assumption of $S=0$ across all traits, for the parameter S that describes MAF-dependent architecture. We agree that it is better to modify the prior distribution, but our choice of gaussian distribution as a prior is strongly dictated by mathematical convenience – other prior distributions might not be practical to work with causal mixture model.

>The MiXeR model tends to clump variants together that are correlated with each other I presume, not 'located too close to each other' as stated.

RESPONSE. We revised the text and replaced “located too close to” with “in high LD with”

> In the inequality under equation (9), the sqrt(k) should be sqrt(K)

RESPONSE. We corrected the typo in the formula.

>The claim made above formula (10) is an interesting one - it would be nice to see whether MiXeR can indeed predict future GWAS discoveries well by using past iterations of GWASs of different sample size on a trait (as is available for numerous traits). I suggest this as optional revision, but it would be nice addition to the paper (not just in the Methods section).

RESPONSE. We plan to explore the power plot of future GWAS discovery in more detail in a separate paper with main focus on a single-trait analysis.

> Equation (10) includes an empty upper limit for the integration that was not added (repeated in the paragraph below).

RESPONSE. There seems to be a technical limitation of the “Equation tools” in MS Word - we have now replaced the upper integration limit with a space. The formula now appears correctly.

> The English in the Methods section should be checked more thoroughly as there are many spelling and grammatical errors.

RESPONSE. We thank the reviewer for pointing this out. We thoroughly checked the manuscript and corrected many spelling and grammatical errors.

We would also like to highlight changes that we made in response to the other reviewer:

1. We extended our analysis to four autoimmune diseases: rheumatoid arthritis (RA), inflammatory bowel disease (IBD), ulcerative colitis (UC), Crohn's disease (CD); and three new anthropomorphic traits: birth weight, waste hip ratio (WHR), body mass index (BMI), in addition to height which was already covered in our previous analysis. The new results are presented in:
 - Suppl. Figures 13h, 13i – Venn diagrams and conditional QQ plots
 - Suppl. Figures 14h, 14i – Observed and estimated bivariate density of z scores
 - Suppl. Figure 15 – projected power of future GWAS
 - Suppl. Figure 16 – univariate QQ plots
 - Suppl. Figures from 17h to 16n – partitioned QQ plots into MAF and LD bins
 - Suppl. Table 6 – meta-data about GWAS studies
 - Suppl. Tables 7, 9 – univariate and bivariate parameter estimates

2. We incorporated Akaike (AIC) and Bayesian (BIC) information criteria into the MiXeR model to validate whether causal mixture has a better fit than infinitesimal model and include the results in Supplementary Table 9.

References

1. Zeng, J. *et al.* Signatures of negative selection in the genetic architecture of human complex traits. *Nature Genetics* **50**, 746-753 (2018).
2. Ruderfer, D.M. *et al.* Genomic Dissection of Bipolar Disorder and Schizophrenia, Including 28 Subphenotypes. *Cell* **173**, 1705-1715.e16 (2018).
3. Bansal, V. *et al.* Genome-wide association study results for educational attainment aid in identifying genetic heterogeneity of schizophrenia. *Nature Communications* **9**, 3078 (2018).
4. Bulik-Sullivan, B.K. *et al.* LD Score regression distinguishes confounding from polygenicity in genome-wide association studies. *Nat Genet* **47**, 291-5 (2015).
5. Zhang, Y., Qi, G., Park, J.-H. & Chatterjee, N. Estimation of complex effect-size distributions using summary-level statistics from genome-wide association studies across 32 complex traits. *Nature Genetics* **50**, 1318-1326 (2018).

Reviewer #1 (Remarks to the Author):

The authors addressed my points, and included autoimmune and morphological trait comparison. I don't have further comments.

Reviewer #2 (Remarks to the Author):

I am satisfied extensive revisions and responses from the authors and only have one remaining comment that I would like them to address:

In the second paragraph of the introduction the authors define 'genetic overlap' (in contrast to 'genetic correlation'), which I think is good to do - but they should change this to 'polygenic overlap' since throughout the paper, and in the abstract, they refer to polygenic overlap not genetic overlap. Also, in this definition it would be helpful to explain whether this proportion is a fraction of the total causal variants across the two traits considered or a fraction of all SNPs in the genome (ie. if 9k are shared among 12k causal variants, is that a polygenic overlap of 0.75 or is it $9k / \text{total number of SNPs}$?)

Below we provide a point-by-point response indicating the changes made to the manuscript. For clarity, our response is written in blue.

REVIEWERS' COMMENTS:

Reviewer #1 (Remarks to the Author):

The authors addressed my points, and included autoimmune and morphological trait comparison. I don't have further comments.

We thank the reviewer for the positive comments and finding our paper of high interest.

Reviewer #2 (Remarks to the Author):

I am satisfied extensive revisions and responses from the authors and only have one remaining comment that I would like them to address:

In the second paragraph of the introduction the authors define 'genetic overlap' (in contrast to 'genetic correlation'), which I think is good to do - but they should change this to 'polygenic overlap' since throughout the paper, and in the abstract, they refer to polygenic overlap not genetic overlap. Also, in this definition it would be helpful to explain whether this proportion is a fraction of the total causal variants across the two traits considered or a fraction of all SNPs in the genome (ie. if 9k are shared among 12k causal variants, is that a polygenic overlap of 0.75 or is it 9k / total number of SNPs?)

We thank the reviewer for the positive comments and finding our paper of high interest. We now renamed "genetic overlap" to "polygenic overlap". We also revised the definition of the polygenic overlap as follows: "For a pair of traits, *polygenic overlap* refers to the fraction of genetic variants causally associated with both traits over the total number of causal variants across the two traits considered."

** See Nature Research's author and referees' website at www.nature.com/authors for information about policies, services and author benefits